# Online Multi-Label Streaming Feature Selection Based on Label Group Correlation and Feature Interaction

**DOI:** 10.3390/e25071071

**Published:** 2023-07-17

**Authors:** Jinghua Liu, Songwei Yang, Hongbo Zhang, Zhenzhen Sun, Jixiang Du

**Affiliations:** 1Department of Computer Science and Technology, Huaqiao University, Xiamen 361021, China; liujinghua@hqu.edu.cn (J.L.); sunway@stu.hqu.edu.cn (S.Y.); zhenzhen_sun@foxmail.com (Z.S.); jxdu@hqu.edu.cn (J.D.); 2Xiamen Key Laboratory of Computer Vision and Pattern Recognition, Huaqiao University, Xiamen 361021, China; 3Fujian Key Laboratory of Big Data Intelligence and Security, Huaqiao University, Xiamen 361021, China

**Keywords:** multi-label feature selection, label group correlation, streaming features, mutual information

## Abstract

Multi-label streaming feature selection has received widespread attention in recent years because the dynamic acquisition of features is more in line with the needs of practical application scenarios. Most previous methods either assume that the labels are independent of each other, or, although label correlation is explored, the relationship between related labels and features is difficult to understand or specify. In real applications, both situations may occur where the labels are correlated and the features may belong specifically to some labels. Moreover, these methods treat features individually without considering the interaction between features. Based on this, we present a novel online streaming feature selection method based on label group correlation and feature interaction (OSLGC). In our design, we first divide labels into multiple groups with the help of graph theory. Then, we integrate label weight and mutual information to accurately quantify the relationships between features under different label groups. Subsequently, a novel feature selection framework using sliding windows is designed, including online feature relevance analysis and online feature interaction analysis. Experiments on ten datasets show that the proposed method outperforms some mature MFS algorithms in terms of predictive performance, statistical analysis, stability analysis, and ablation experiments.

## 1. Introduction

Multi-label feature selection (MFS) plays a crucial role in addressing the preprocessing of high-dimensional multi-label data. Numerous methods have been proposed and proven to be effective in improving prediction performance and model interpretability. However, traditional MFS methods assume that all features are collected and presented to the learning model beforehand [1,2,3,4]. This assumption does not align with many practical application scenarios where not all features are available in advance. In video recognition, for example, each frame may possess important features that become available over time. Hence, achieving real-time feature processing has emerged as a significant concern [5,6,7,8].

Online multi-label feature selection with streaming features is an essential branch of MFS that facilitates the efficient real-time management of streaming features. It provides significant advantages, such as low time and space consumption, particularly when dealing with extremely high-dimensional datasets. Some notable works in this have attracted attention, including online multi-label streaming feature selection based on a neighborhood rough set (OM-NRS) [9], multi-label streaming feature selection (MSFS) [10], and novel streaming feature selection(ASFS) [11]. However, these methods primarily focus on eliminating irrelevant and/or redundant features. In addition to identifying irrelevant and/or redundant features, feature interaction is crucial but often overlooked. Feature interaction refers to features that have weak or independent correlations with the label, but when combined with other features, they may exhibit a strong association with the predicted label. Streaming feature selection with feature interaction (SFS-FI) [12] is a representative approach that considers feature interaction dynamically. SFS-FI has successfully identified the impact of feature interaction; however, it lacks the capability to tackle the learning challenge in multi-label scenarios.

Another difficulty with online MFS is that labels are universally correlated, which is a distinctive property of multi-label data [13,14,15,16]. Intuitively, a known label can aid in learning an unknown one, and the co-occurrence of two labels may provide additional information to the model. For example, an image with ‘grassland’ and ‘lion’ labels is likely also to be marked as ‘African’; similarly, ‘sea’ and ‘ship’ labels would tend to appear together in a short video, while ‘train’ and ‘sea’ labels tend not to appear together. Some research work has been carried out around label correlation. Representative work includes multi-label streaming feature selection (MSFS) and online multi-label streaming feature selection with label correlation ($OMSFS_{LC}$). MSFS captures label correlation by constructing a new data representation pattern for label space and utilizes the constructed label relationship matrix to examine the merits of features. $OMSFS_{LC}$ constructs label weights by calculating label correlation, and, on this basis, integrates label weights into significance analysis and relevance analysis of streaming features. The methods mentioned above select features by evaluating the relationship between features and the global label space. This strategy may not be optimal as it is challenging to comprehend and specify the relationship between relevant labels and features. Based on research by Li et al. [17], it has been found that strongly correlated labels tend to share similar specific features, while weakly related labels typically have distinct features. In line with this observation, this paper will categorize related labels into two groups: strongly related labels will be grouped together, while weakly related labels will be separated into different groups.

Accordingly, a novel online multi-label streaming feature selection based on label group correlation and feature interaction, namely OSLGC, is proposed to select pertinent and interactive features from streaming features. Firstly, our method involves calculating the correlation matrix of labels and using graph theory to group related labels. Labels within the same group have a strong correlation, while labels from different groups have a weak correlation. Then, we define the feature relevance item and integrate the label weight and feature interaction weight into the feature relevance item. Subsequently, a framework based on sliding windows is established, which iteratively processes streaming features through two steps: online feature correlation analysis and online feature interaction analysis. Finally, extensive experiments demonstrate that OSLGC can yield significant performance improvements compared to other mature MFS methods. The uniqueness of OSLGC is established as follows:By utilizing graph theory, label groups are constructed to ensure that closely associated labels are grouped together. This method provides an effective means of visualizing the relationships among labels.We provide a formal definition of feature interaction and quantify the impact of feature interaction under different label groups. Based on this, OSLGC is capable of selecting features with interactivity.A novel streaming feature selection framework using sliding windows is proposed, which resolves the online MFS problem by simultaneously taking feature interaction, label importance, as well as label group correlation, into account.Experiments on ten datasets demonstrate that the proposed method is competitive with existing mature MFS algorithms in terms of predictive performance, statistical analysis, stability analysis, and ablation experiments.

The rest of this article is arranged as follows: In Section 2, we review previous research. Section 3 provides the relevant preparatory information. In Section 4, we present the detailed procedure for OSLGC. In Section 5, we report the empirical study. Finally, Section 6 sums up the work of this paper and considers the prospects, priorities and direction of future research.

## 2. Related Work

Multi-label feature selection (MFS), as a widely known data preprocessing method, has achieved promising results in different application fields, such as emotion classification [18], text classification [19], and gene detection [20]. Depending on whether the features are sufficiently captured in advance, existing MFS methods can be divided into batch and online methods.

The batch method assumes that the features presented to learning are pre-available. Generally speaking, it can be further subdivided into several types according to the characteristics provided by the complex label space, including missing labels [21,22], label distribution [23,24], label selection [25], label imbalance [26,27], streaming labels [28,29], partial labels [30,31], label-specific features [32,33,34], and label correlation [35,36,37]. Among them, investigating label correlation is considered to be a favorable strategy to promote the performance of learning. To date, many works have focused on this. For instance, label supplementation for multi-label feature selection (LSMFS) [38] evaluates the relationship between labels using mutual information provided by the features. Quadratically constrained linear programming (QCLP) [39] introduces a matrix variable normal prior distribution to model label correlation. By minimizing the label ranking loss of label correlation regularization, QCLP is able to identify a feature subset. On the other hand, label-specific features emphasize that different labels may possess their own specific features. One of the most representative studies, Label specIfic FeaTures (LIFT) [32], has shown that using label-specific features to guide the MFS process can elevate the performance and interpretability of learning tasks. Recently, group-preserving label-specific feature selection (GLFS) [33] has been used to exploit label-specific features and common features with l_2,1_-norm regularization to support the interpretability of the selected features.

The online method differs from the batch method in that features are generated on-the-fly and feature selection takes place in real-time as the features arrive. Based on the characteristics of the label space, it can be categorized into two groups: label independence and label correlation. For label independence, several methods have been proposed, such as the streaming feature selection algorithm with dynamic sliding Windows and feature repulsion loss (SF-DSW-FRL) [40], multi-objective online streaming multi-label feature selection using mutual information (MOML) [41], streaming feature selection via class-imbalance aware rough set (SFSCI) [42], online multi-label group feature selection (OMGFS) [43], and multi-objective multi-label-based online feature selection (MMOFS) [44]. Similar to the static MFS methods, the online MFS approach also focuses on exploring label correlation. For instance, MSFS [10] uses the relationship between samples and labels to construct a new data representation model to measure label correlation, and implements feature selection by designing feature correlation and redundancy analysis. Multi-label online streaming feature selection with mutual information (ML-OSMI) [45] uses high-order methods to determine label correlation, and combines spectral granulation and mutual information to evaluate streaming features. Unfortunately, existing methods cannot exactly capture the impact of label relationships on the evaluation of streaming features and are hindered by a time-consuming calculation procedure. Thus, online multi-source streaming features selection (OMSFS) [7] investigates label correlation by calculating mutual information, and, on this basis, constructs the weight for each label and designs a significance analysis to accelerate the computational efficiency.

Based on our review of previous studies, we find that with the arrival of each new feature, existing methods can be effective in dealing with streaming features. However, these methods pay more attention to the contribution of features to all labels, and do not explore the specific relationship between features and labels. To put it simply, they fail to consider that highly correlated labels may have common features, while weakly correlated labels may have distinct features. Additionally, it is important to mention that most previous works have focused on selecting the most relevant features to labels, but have ignored the potential contribution of feature interactions to labels. In contrast, our framework pays close attention to feature interactions and label group correlation, and seeks to explore the specific features and label group weights corresponding to the label group.

## 3. Preliminaries

### 3.1. Multi-Label Learning

Given a multi-label information table MLS=<U,F,L>, where $U=\{x_{1},x_{2},\cdots ,x_{n}\}$ is a non-empty instance set, $F=\{f_{1},f_{2},\cdots ,f_{d}\}$ and $L=\{l_{1},l_{2},\cdots ,l_{m}\}$ are a feature set and label set used to describe instances, respectively. $l_{i}\left(x_{k}\right)$ represents the value of label $l_{i}$ on instance $x_{k}\in U$, where $l_{i}\left(x_{k}\right)=1$, only if $x_{k}$ possesses $l_{i}$, and 0 otherwise. The task of multi-label learning is to learn a function $h:U\to 2^{L}$.

### 3.2. Basic Information-Theoretic Notions

This section introduces some basic information theory concepts which are commonly used in the evaluation of feature quality.

**Definition** **1.**
*Let $X=\{x_{1},x_{2},\cdots ,x_{n}\}$ be a discrete random variable and $P(x_{i})$ be the probability of $x_{i}$, then the entropy of X is*

(1)
$$\begin{array}{c} H\left(X\right)=-\sum_{x_{i}\in X}P\left(x_{i}\right)logP\left(x_{i}\right). \end{array}$$


*H(X) is a measure of randomness or uncertainty in the distribution of X. It is at a maximum when all the possible values of X are equal, and at a minimum when X takes only one value with probability 1.*


**Definition** **2.**
*Let $Y=\{y_{1},y_{2},\cdots ,y_{m}\}$ be another random variable. Then, the joint entropy H(X,Y) of X and Y is:*

(2)
$$\begin{array}{c} H\left(X,Y\right)=-\sum_{x_{i}\in X}\sum_{y_{j}\in Y}P\left(x_{i},y_{j}\right)logP\left(x_{i},y_{j}\right), \end{array}$$

*where $P(x_{i},y_{j})$ denotes the joint probability of $x_{i}$ and $y_{j}$.*


**Definition** **3.**
*Given X and Y, when the variable Y is known, the residual uncertainty of X can be determined by the conditional entropy H(X|Y):*

(3)
$$\begin{array}{c} H\left(X|Y\right)=-\sum_{x_{i}\in X}\sum_{y_{j}\in Y}P\left(x_{i},y_{j}\right)logP\left(x_{i}|y_{j}\right), \end{array}$$

*where $P(x_{i}|y_{j})$ is the conditional probability of $x_{i}$ given $y_{j}$.*


**Definition** **4.**
*Given X and Y, then the amount of information shared by two variables can be determined by the mutual information I(X;Y):*

(4)
$$\begin{array}{c} I\left(X;Y\right)=\sum_{x_{i}\in X}\sum_{y_{j}\in Y}P\left(x_{i},y_{j}\right)log\frac{P(y_{j}|x_{i})}{P(y_{j})}. \end{array}$$


*The larger the I(X;Y) value, the stronger the correlation between the two variables. Inversely, the two variables are independent if I(X;Y)=0.*


**Definition** **5.**
*Given variables X, Y and Z, when Z is given, the uncertainty of X due to the known Y can be measured by the conditional mutual information I(X;Y∣Z):*

(5)
$$\begin{array}{c} I\left(X;Y|Z\right)=\sum_{x_{i}\in X}\sum_{y_{j}\in Y}\sum_{z_{k}\in Z}P\left(x_{i},y_{j},z_{k}\right)\log\frac{P(x_{i}|y_{j},z_{k})}{P(x_{i}|y_{j})}. \end{array}$$



## 4. The OSLGC Method

### 4.1. Exploiting Label Group Correlation

To investigate label group correlation, in this subsection, we introduce a graph-based method to further distinguish the relevant labels, which can effectively differentiate relevant labels by grouping strongly related labels together and separating weakly related ones into different groups. The process involves two fundamental steps: (1) constructing an undirected graph of the labels, and (2) partitioning the graph to create distinct label groups.

In the first step, OSLGC aims to construct an undirected graph that effectively captures the label correlation among all labels, thus providing additional information for streaming feature evaluation. For this purpose, it is necessary to investigate the correlation between labels.

**Definition** **6.**
*Given <U,F,L>, $x_{k}\in U$, $l_{i},l_{j}\in L$, $l_{i}\left(x_{k}\right)$ represents the value of label $l_{i}$ with respect to instance $x_{k}$, the correlation $r_{ij}$ between the labels is defined as:*

(6)
$$\begin{array}{c} r_{ij}=\sum_{x_{k}\in U}P\left(l_{i}\left(x_{k}\right),l_{j}\left(x_{k}\right)\right)\log\frac{P(l_{i}\left(x_{k}\right),l_{j}\left(x_{k}\right))}{P\left(l_{i}\left(x_{k}\right)\right)P\left(l_{j}\left(x_{k}\right)\right)}. \end{array}$$


*Obviously, if $l_{i}$ and $l_{j}$ are independent, then $r_{ij}=0$; otherwise, $r_{ij}>0$.*


Using Equation (6), the label correlation matrix $M(R_{LC})$ is obtained, and the form of $M(R_{LC})$ is shown below.
$$M\left(R_{LC}\right)=\left[\begin{array}{cccc} r_{11} & r_{12} & \cdots & r_{1m}\\ r_{21} & r_{22} & \cdots & r_{2m}\\ \vdots & \vdots & \vdots & \vdots \\ r_{m1} & r_{m2} & \cdots & r_{mm} \end{array}\right].$$

Based on the matrix, the weighted undirected graph of the label correlation can be structured Graph=(V,E), where $V=\{l_{i}|l_{i}\in L\wedge i\in \left[1,m\right]\}$ and $E=\{\left(l_{i},l_{j}\right)|l_{i},l_{j}\in L,$i,j∈[1,m],i≠j} mean the vertex and edge of Graph, respectively. As $M(R_{LC})$ is symmetric, Graph is an undirected graph that reflects the correlation among all labels. But, regrettably, Graph has *m* vertices and m(m−1)/2 edges. For ultra-high-dimensional data, the density of the graph will be considerable, which often leads to strong edge interweaving of different weights. Moreover, the resolution of complete graphs is an NP-hard problem. Therefore, for Graph, it is necessary to reduce the edges of Graph.

In the second step, OSLGC aims to divide the graph and create label groups. With this intent, we first generate a minimum spanning tree (MST) through the Prim algorithm. MST has the same vertices as Graph and  partial edges of Graph. The weight of the link edge in the MST is expressed as $W_{\left(l_{i},l_{j}\right)}$, which is essentially different for different edges. To divide strongly correlated labels into groups, we set the threshold to break the edges below the threshold in MST.

**Definition** **7.**
*Given $W_{\left(l_{i},l_{j}\right)}$ represents the weight of the edges, and the threshold for weak label correlation is defined as:*

(7)
$$\begin{array}{c} \delta =\frac{\sum_{(l_{i},l_{j})\in MST}W_{\left(l_{i},l_{j}\right)}}{m-1}. \end{array}$$


*δ is the average of the edge weights, which is used to divide the label groups, thereby putting the strongly related labels in the same group.*

*Concretely, if $W_{\left(l_{i},l_{j}\right)}\geq \delta$, which means that the relationship between labels $l_{i}$ and $l_{j}$ is a strength label correlation, we will reserve the edge that connects $l_{i}$ with $l_{j}$. If $W_{\left(l_{i},l_{j}\right)}<\delta$, which explains that the relationship between labels $l_{i}$ and $l_{j}$ is a weakness label correlation, we can remove the edge that connects $l_{i}$ with $l_{j}$ from MST. Hence, the MST can be segmented into forests by threshold segmentation. In the forest, the label nodes within each subtree are strongly correlated, while the label nodes between subtrees are weakly correlated. Based on this, we can treat each subtree as a label group, denoted as $L=\{LG_{1}\cup LG_{2}\cup \cdots \cup LG_{p}\}$.*


**Example** **1.**
*A multi-label dataset is presented in Table 1. First, the label correlation matrix is calculated using Equation (6), as follows:*

$$M\left(R_{LC}\right)=\left[\begin{array}{cccccc} 1.00 & 0.13 & 0.05 & 0.61 & 0.02 & 0.26\\ 0.13 & 1.00 & 0.13 & 0.28 & 0.00 & 0.13\\ 0.05 & 0.13 & 1.00 & 0.00 & 0.02 & 0.02\\ 0.61 & 0.28 & 0.00 & 1.00 & 0.00 & 0.13\\ 0.02 & 0.00 & 0.02 & 0.00 & 1.00 & 0.26\\ 0.26 & 0.13 & 0.02 & 0.13 & 0.26 & 1.00 \end{array}\right].$$



Then, we can create the label undirected graph by using the label correlation matrix, as shown in Figure 1a. Immediately afterwards, the minimum spanning tree is generated by the Prim algorithm, as shown in Figure 1b. Finally, the threshold δ of *MST* is calculated using Equation (7), and the edges that meet condition $W_{\left(l_{i},l_{j}\right)}<\delta$ are removed, as shown in Figure 1c.

### 4.2. Analysis Feature Interaction under Label Group

As a rule, the related labels generally share some label-specific features [17,33], i.e., labels within the same label group may share the same label-specific features. Thus, to generate label-specific features for different label groups, in this subsection, we will further analyze feature relationships under different label groups, including feature independence, feature redundancy, and feature interaction. We also give the interaction weight factor to quantify the influence degree of the feature relationship under different label groups.

**Definition** **8**(Feature independence)**.** *Given a set of label groups $L=\{LG_{1}\cup LG_{2}\cup \cdots \cup LG_{p}\}$, $LG_{h}\subseteq L$, $S_{t}=\left\{f_{1},f_{2},\cdots ,f_{d^{*}}\right\}$ denotes the selected features, and $f_{t}$ is a new incoming feature at time t. For $\forall f_{i}\in S_{t}$, $f_{i}$ and $f_{t}$ are referred to as feature independence under ${LG}_{h}$ if, and only if:*
(8)$$\begin{array}{c} I\left(f_{i};LG_{h}\right)+I\left(f_{t};LG_{h}\right)=I\left(f_{i},f_{t};LG_{h}\right). \end{array}$$

According to Definition 8, $I\left(f_{i};LG_{h}\right)+I\left(f_{t};LG_{h}\right)=I\left(f_{i},f_{t};LG_{h}\right)$ suggests that the information provided by feature $f_{i}$ and $f_{t}$ for the label group $LG_{h}$ are non-interfering, i.e., the features are independent of each other under label group $LG_{h}$.

**Theorem** **1.**
*If $I\left(f_{t};{LG}_{h}|f_{i}\right)=I\left(f_{t};{LG}_{h}\right)$ or $I\left(f_{i};{LG}_{h}|f_{t}\right)=I\left(f_{i};{LG}_{h}\right)$, then $f_{i}$ and $f_{t}$ are independent under label group $LG_{h}$.*


**Proof.** $I\left(f_{i},f_{t};LG_{h}\right)=I\left(f_{i};LG_{h}\right)+I\left(f_{t};LG_{h}|f_{i}\right)$ = $I(f_{t};LG_{h})$ + $I(f_{i};LG_{h}|f_{t})$. If $I\left(f_{i};LG_{h}|f_{t}\right)=I\left(f_{i};LG_{h}\right)$ or $I\left(f_{t};LG_{h}|f_{i}\right)=I\left(f_{t};LG_{h}\right)$, $I\left(f_{i},f_{t};LG_{h}\right)=I\left(f_{i};LG_{h}\right)+I\left(f_{t};LG_{h}\right)$. Thus, $f_{i}$ and $f_{t}$ are independent under label group $LG_{h}$.    □

**Theorem** **2.**
*If $f_{i}$ and $f_{t}$ are independent, under the condition that label group $LG_{h}$ is known, then $I(f_{i};f_{t}|LG_{h})=0$.*


**Proof.** If $f_{i}$ and $f_{t}$ are independent, i.e., $I(f_{i};f_{t})=0$, according to Definition 5, it can be proven that $I(f_{i};f_{t}|LG_{h})=0$.   □

**Definition** **9**(Feature redundancy)**.** *Given a set of label groups $L=\{LG_{1}\cup LG_{2}\cup \cdots \cup LG_{p}\}$, $LG_{h}\subseteq L$, $S_{t}=\left\{f_{1},f_{2},\cdots ,f_{d^{*}}\right\}$ denotes the selected features, and $f_{t}$ is a new incoming feature. For $\forall f_{i}\in S_{t}$, $f_{i}$ and $f_{t}$ are referred to as feature redundancy under ${LG}_{h}$ if, and only if:*
(9)$$\begin{array}{c} I\left(f_{i};LG_{h}\right)+I\left(f_{t};LG_{h}\right)>I\left(f_{i},f_{t};LG_{h}\right). \end{array}$$

Equation (9) suggests that there is partial duplication of information provided by two features; that is, the amount of information brought by two features $f_{i}$ and $f_{t}$ together for label group ${LG}_{h}$ is less than the sum of the information brought by the two features for ${LG}_{h}$ separately.

**Theorem** **3.**
*If $I\left(f_{i};LG_{h}|f_{t}\right)<I\left(f_{i};LG_{h}\right)$ or $I\left(f_{t};LG_{h}|f_{i}\right)<I\left(f_{t};LG_{h}\right)$, then the relationship between $f_{i}$ and $f_{t}$ is a pair of feature redundancy under label group $LG_{h}$.*


**Proof.** $I\left(f_{i},f_{t};LG_{h}\right)=I\left(f_{i};LG_{h}\right)+I\left(f_{t};LG_{h}|f_{i}\right)$ = $I(f_{t};LG_{h})$ + $I(f_{i};LG_{h}|f_{t})$. If $I\left(f_{i};LG_{h}|f_{t}\right)<I\left(f_{i};LG_{h}\right)$ or $I\left(f_{t};LG_{h}|f_{i}\right)<I\left(f_{t};LG_{h}\right)$, $I\left(f_{i},f_{t};LG_{h}\right)<I\left(f_{i};LG_{h}\right)+I\left(f_{t};LG_{h}\right)$. Thus, the relationship between $f_{i}$ and $f_{t}$ is a pair of feature redundancy under label group $LG_{h}$.    □

**Definition** **10**(Feature interaction)**.** *Given a set of label groups $L=\{LG_{1}\cup LG_{2}\cup \cdots \cup LG_{p}\}$, $LG_{h}\subseteq L$, $S_{t}=\left\{f_{1},f_{2},\cdots ,f_{d^{*}}\right\}$ denotes the selected features, and $f_{t}$ is a new incoming feature. For $\forall f_{i}\in S_{t}$, $f_{i}$ and $f_{t}$ are referred to as a feature interaction under ${LG}_{h}$ if, and only if:*
(10)$$\begin{array}{c} I\left(f_{i};LG_{h}\right)+I\left(f_{t};LG_{h}\right)<I\left(f_{i},f_{t};LG_{h}\right). \end{array}$$

Equation (10) suggests that there is a synergy between features $f_{i}$ and $f_{t}$ together for label group ${LG}_{h}$; that is, they yield more information together for label group ${LG}_{h}$ than what could be expected from the sum of $I(f_{i};LG_{h})$ and $I(f_{t};LG_{h})$.

**Theorem** **4.**
*If $I\left(f_{i};LG_{h}|f_{t}\right)>I\left(f_{i};LG_{h}\right)$ or $I\left(f_{t};LG_{h}|f_{i}\right)>I\left(f_{t};LG_{h}\right)$, then $f_{i}$ and $f_{t}$ is a pair of feature interaction under label group $LG_{h}$.*


**Proof.** $I\left(f_{i},f_{t};LG_{h}\right)=I\left(f_{i};LG_{h}\right)+I\left(f_{t};LG_{h}|f_{i}\right)=I\left(f_{t};LG_{h}\right)+I\left(f_{i};LG_{h}|f_{i}\right)$. If $I(f_{i};LG_{h}|f_{t})$$>I(f_{i};LG_{h})$ or $I\left(f_{t};LG_{h}|f_{i}\right)>I\left(f_{t};LG_{h}\right)$, $I\left(f_{i},f_{t};LG_{h}\right)>I\left(f_{i};LG_{h}\right)+I\left(f_{t};LG_{h}\right)$. Thus, $f_{i}$ and $f_{t}$ are a pair of feature positive interaction under label group $LG_{h}$.    □

**Property** **1.**
*If two features*

$f_{i}$

*and*

$f_{t}$

*are not independent, the correlations between*

$f_{i}$

*and*

$f_{t}$

*under a different label group*

$LG_{h}$

*are distinct. It is easy to show with Example 2.*


**Example** **2.**
*Continue Table 1. As shown in Table 2, we can see that $I(f_{1},f_{2};LG_{1})=0.997$ is less than $I\left(f_{1};LG_{1}\right)+I\left(f_{2};LG_{1}\right)=1.227$, and, according to Definition 9, $f_{1}$ and $f_{2}$ is a feature redundancy under label group $LG_{1}$. However, for label group $LG_{3}$, it satisfies that $I\left(f_{1},f_{2};LG_{3}\right)>I\left(f_{1};LG_{3}\right)+I\left(f_{2};LG_{3}\right)$; that is, $f_{1}$ and $f_{2}$ is a feature interaction under the label group $LG_{3}$. This finding suggests that the relationship between $f_{1}$ and $f_{2}$ changes dynamically under different label groups.*

*Consequently, to evaluate features accurately, it is imperative to quantify the influence of the feature relationships on feature relevance. That is, the inflow of a new feature $f_{t}$ has a positive effect in predicting labels, and we should enlarge the weight of feature $f_{t}$; otherwise, the weight of feature $f_{t}$ should be reduced. The feature interaction weight factor is defined to quantize the impact of the feature relationships as follows:*


**Table 2 entropy-25-01071-t002:** The relationship between features.

Mutual Information	Combination	Feature Relationship
$I(f_{1};LG_{1})=0.771$	$I(f_{1},f_{2};LG_{1})=0.997$	Feature redundancy
$I(f_{2};LG_{1})=0.446$		
$I(f_{1};LG_{3})=0.020$	$I(f_{1},f_{2};LG_{3})=0.171$	Feature interaction
$I(f_{2};LG_{3})=0.020$		

**Definition** **11**(Feature Interaction Weight)**.** *Given a set of label groups $L=\{LG_{1}\cup LG_{2}\cup \cdots \cup LG_{p}\}$, $LG_{h}\subseteq L$,$S_{t}=\left\{f_{1},f_{2},\cdots ,f_{d^{*}}\right\}$ denotes the selected features, and $f_{t}$ is a new incoming feature. For $\forall f_{i}\in S_{t}$, the feature interaction weight between $f_{i}$ and $f_{t}$ is defined as:*
(11)$$\begin{array}{c} FW\left(f_{i},f_{t},LG_{h}\right)=\frac{I(f_{i},f_{t};LG_{h})}{I\left(f_{i};LG_{h}\right)+I\left(f_{t};LG_{h}\right)}. \end{array}$$

$FW(f_{i},f_{t},LG_{h})$ offers additional information for evaluating feature $f_{t}$. If feature $f_{t}$ and the selected feature $f_{i}\in S_{t}$ is independent or redundant, it holds that $FW(f_{i},f_{t},LG_{h})\leq 1$. However, if the feature relationship is interactive, it holds that $FW(f_{,}f_{t},LG_{h})>1$.

### 4.3. Streaming Feature Selection with Label Group Correlation and Feature Interaction

Streaming features refer to features acquired over time; however, in fact, not all features obtained dynamically are helpful for prediction. Therefore, it is necessary to extract valuable features from the streaming features’ environment. To achieve this purpose, in this paper, we implement the analysis of streaming features in two stages: online feature relevance analysis and online feature interaction analysis.

#### 4.3.1. Online Feature Relevance Analysis

The purpose of feature relevance analysis is to select features that are important to the label groups. Correspondingly, the feature relevance is defined as follows:

**Definition** **12**(Feature Relevance)**.** *Given label groups $L=\{LG_{1}\cup LG_{2}\cup \cdots \cup LG_{p}\}$, $f_{t}$ is a new incoming feature, the feature relevance item is defined as:*
(12)$$\begin{array}{c} \gamma \left(f_{t}\right)=\sum_{h=1}^{p}I(f_{t};LG_{h})\times W\left(LG_{h}\right). \end{array}$$

In which, $W(LG_{h})$ denotes the weight assigned to each label group, and $W\left(LG_{h}\right)=\frac{H(LG_{h})}{\sum_{j=1}^{p}H(LG_{j})}$ where $H(LG_{h})$ is the information entropy of label group $LG_{h}$. The higher the weight of the label group, the more important the label group is to other label groups. In other words, the corresponding label-specific features of the label group should have higher feature importance.

**Definition** **13.**
*Given label groups $L=\{LG_{1}\cup LG_{2}\cup \cdots \cup LG_{p}\}$, $f_{t}$ is a new incoming feature, and $\gamma (f_{t})$ is the feature relevance. With a pair of thresholds α and β (0<α<β), we define:*

*(1) $f_{t}$ is strongly relevant, if $\beta \leq \gamma (f_{t})$;*

*(2) $f_{t}$ is weakly relevant, if $\alpha <\gamma (f_{t})<\beta$;*

*(3) $f_{t}$ is irrelevant, if $0\leq \gamma (f_{t})\leq \alpha$.*


In general, for a new incoming feature $f_{t}$, if $f_{t}$ is powerfully relevant, we will select it; if $f_{t}$ is irrelevant, we will directly abandon it and no longer consider it later; if $f_{t}$ is weakly relevant, there is a risk of greater misjudgment in making a decision immediately, including selecting or abandoning, and the best approach is to obtain more information to make a decision.

#### 4.3.2. Online Feature Interaction Analysis

Definition 13 can be used to make intuitive judgments about features that are weakly correlated. However, Definition 13 does not provide a basis for selecting or abandoning weakly relevant features. Therefore, it is necessary to determine whether to remove or retain the weakly relevant features.

**Definition** **14.**
*Given label groups $L=\{LG_{1}\cup LG_{2}\cup \cdots \cup LG_{p}\}$, $S_{t}=\left\{f_{1},f_{2},\cdots ,f_{d^{*}}\right\}$ denotes the selected features, and $f_{t}$ is a new incoming feature. The feature relevance when considering feature interaction, called the enhanced feature relevance, is defined as:*

(13)
$$\begin{array}{c} 𝟊\left(f_{t}\right)=\frac{1}{|S_{t}|}\sum_{i=1}^{d^{*}}\sum_{h=1}^{p}I(f_{t};LG_{h})\times W\left(LG_{h}\right)\times FW\left(f_{i},f_{t};LG_{h}\right). \end{array}$$



In which, $FW(f_{i},f_{t};LG_{h})$ is the feature interaction weight between $f_{t}$ and $f_{i}\in S_{t}$. Furthermore, to determine whether to retain the weakly relevant feature, we set the mean value of feature relevance about the selected features as the relevance threshold, as follows:

**Definition** **15.**
*Given $S_{t}=\left\{f_{1},f_{2},\cdots ,f_{d^{*}}\right\}$ denotes the selected features, $f_{i}\in S_{t}$, at time t, the mean value of the feature relevance about the selected features is:*

(14)
$$\begin{array}{c} Mean_{t}=\sum_{i=1}^{d^{*}}\frac{\gamma (f_{i})}{|S_{t}|}. \end{array}$$



Obviously, when $𝟊\left(f_{t}\right)>Mean_{t}$, it shows that the weak relevant feature $f_{t}$ interacts with the selected features. In this case, $f_{t}$ can enhance the prediction ability and be selected as an effective feature. Otherwise, when $𝟊\left(f_{t}\right)\leq Mean_{t}$, it denotes that adding the weakly relevant feature $f_{t}$ does not promote the prediction ability for labels, and, in this case, we can discard the feature $f_{t}$.

#### 4.3.3. Streaming Feature Selection Strategy Using Sliding Windows

According to Definition 13, two main issues need to be addressed: (1) how to design a streaming feature selection mechanism to discriminate the newly arrived features; (2) how to set proper thresholds for α and β.

**(1) Streaming feature selection with sliding windows**: To solve the first challenge, a sliding window mechanism is proposed to receive the arrived features in a timed sequence, which is consistent with the dynamic nature of the streaming features. The specific process can be illustrated using the example in Figure 2.
First, the sliding window (SW) continuously receives and saves the arrived features. When the number of features in the sliding window reaches the preset size, the features in the window are discriminated, which includes decision-making with regard to selection, abandonment, or delay.According to the feature relevance $\gamma (f_{t})$ (Definition 12), we select the strongly relevant features, as shown in Figure 2. We can straightforwardly select strongly relevant features, e.g., $f_{15}$ and $f_{18}$. Similarly, for the irrelevant features, they are discarded from the sliding window, e.g., $f_{16}$ and $f_{17}$.For weakly relevant features, we need to further analyze the enhanced feature relevance by considering the feature interaction. If the weakly relevant features satisfy the condition $𝟊\left(f_{t}\right)>Mean_{t}$, they can be selected, e.g., $f_{19}$; otherwise, the weakly relevant features are retained in the sliding window, for example, $f_{14}$, and new features are awaited to flow into the sliding window.

This process is performed repeatedly. That is, when the features in the sliding window reach saturation or no new features appear, the next round of feature analysis is performed.

**Figure 2 entropy-25-01071-f002:**
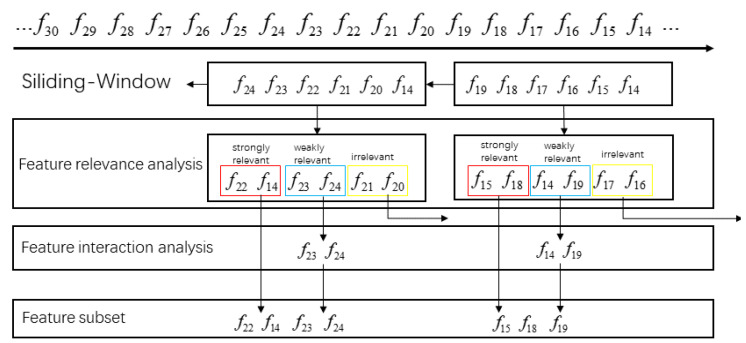
Streaming feature selection with sliding window.

**(2) Thresholds setting of α and β**: To solve the second challenge, we assume that the experimental data follow a normal distribution and the streaming features arrive randomly. Inspired by the 3σ principle of normal distribution, we set α and β as the mean and standard deviation of features in the sliding window.

**Definition** **16.**
*Given a sliding window SW, $f_{t^{{}^{\prime }}}\in SW$, and $\gamma (f_{t^{{}^{\prime }}})$ is the feature relevance, then, at time t, the mean value $\mu_{t}$ of the sliding window is:*

(15)
$$\begin{array}{c} \mu_{t}=\sum_{f_{t^{{}^{\prime }}}\in SW}\frac{\gamma (f_{t^{{}^{\prime }}})}{\left|SW\right|}. \end{array}$$



**Definition** **17.**
*Given a sliding window SW, $f_{t^{{}^{\prime }}}\in SW$, and $\gamma (f_{t^{{}^{\prime }}})$ is the feature relevance, then, at time t, the standard deviation $\sigma_{t}$ of the sliding window is:*

(16)
$$\begin{array}{c} \sigma_{t}=\sqrt{\frac{\sum_{f_{t^{{}^{\prime }}}\in SW}{\left(\gamma \left(f_{t^{{}^{\prime }}}\right)-\mu_{t}\right)}^{2}}{\left|SW\right|}}. \end{array}$$



Therefore, we combine the 3σ principle of normally distributed data to redefine the three feature relationships.

**Definition** **18.**
*Given $\gamma (f_{t})$ is the feature relevance, at time t, $\mu_{t}$ and $\sigma_{t}$ are the mean and standard deviation of the features in the sliding window. Then, we define three feature relationships as:*

*(1) $f_{t}$ is strongly relevant, if $\mu_{t}+\sigma_{t}\leq \gamma \left(f_{t}\right)$;*

*(2) $f_{t}$ is weakly relevant, if $\mu_{t}-\sigma_{t}<\gamma \left(f_{t}\right)<\mu_{t}+\sigma_{t}$;*

*(3) $f_{t}$ is irrelevant, if $0\leq \gamma \left(f_{t}\right)\leq \mu_{t}-\sigma_{t}$.*


Through the above analysis, we propose a novel algorithm, named OSLGC, as shown in Algorithm 1.
**Algorithm 1** The OSLGC algorithm**Input**: SW: sliding window, $f_{i}$: predictive features, *L*: label set.**Output**: $S_{t}$: the feature subset at time *t*.
1:Generate label groups $L=\{LG_{1}\cup LG_{2}\cup \cdots \cup LG_{p}\}$ by Section 4.1;2:**repeat**3:   Get a new feature $f_{t}$ at time *t*;4:   Add feature $f_{t}$ to the sliding window SW;5:   **while** SW is full or no features are available **do**6:     Compute $\mu_{t},\gamma_{t},$ and $Mean_{t}$;7:     **for** each $f_{t^{{}^{\prime }}}\in SW$ **do**8:        **if** $\gamma \left(f_{t^{{}^{\prime }}}\right)\geq \mu_{t}+\sigma_{t}$ **then**9:          **if** $𝟊\left(f_{t^{{}^{\prime }}}\right)>Mean_{t}$ **then**10:             $S_{t}=S_{t}\cup f_{t}^{{}^{\prime }}$;11:          **end if**12:        **else**13:          Discard $f_{t}^{{}^{\prime }}$;14:        **end if**15:     **end for**16:   **end while**17:**until** No features are available;18:Return $S_{t}$;

The major computation in OSLGC is feature analysis in sliding windows (Steps 5–16). Assuming $|F_{t}|$ is the number of currently arrived features, and |L| is the number of labels, in the best-case scenario, OSLGC obtains a feature subset after running online feature relevance analysis, and the time complexity is $O(|F_{t}|\cdot |L|)$. However, in many cases, the features are not simply strongly relevant or irrelevant, but include weakly relevant instances. Therefore, online feature interaction analysis needs to be further performed. The final time complexity is $O(|F_{t}|\cdot |F_{t}|\cdot |L|)$.

## 5. Experiments

### 5.1. Data Sets

We conducted experiments on ten multi-label datasets, which were mainly from three different domains, that is, text, audio, and images, respectively. Among them, the first eight datasets (i.e., Business, Computer, Education, Entertainment, Health, Entertainment, Reference, and Society) were taken from Yahoo, and were derived from the actual web text classification. For audio, Birds is an audio dataset that identifies 19 species of birds. For images, Scene includes 2407 images with up to six labels per image. These datasets are freely available for public download and have been widely used in research on multi-label learning.

Detailed information is provided in Table 3. For each dataset *S*, we use |S|, F(S), and L(S) to represent the number of instances, number of features, and number of labels, respectively. LCard(S) denotes the average number of labels per example, and LDen(S) standardizes LCard(S) according to the number of possible labels. In addition, it is worth noting that the number of instances and the number of labels in different datasets vary from 645 to 5000 and from 6 to 33, respectively. These datasets with varied properties provide a solid foundation for algorithm testing.

### 5.2. Experimental Setting

To visualize the performance of OSLGC, we compared OSLGC with several recent MFS algorithms. For a reasonable comparison, two different types of algorithms were selected as comparison algorithms, including (1) two online multi-label streaming feature selection algorithms, and (2) five MFS methods based on information theory. Specifically, the two online multi-label streaming feature selection methods included multi-label streaming feature selection (MSFS) [10] and online multi-label feature selection based on neighborhood rough set (OMNRS) [9]. On the other hand, the five MFS methods based on information theory were multi-label feature selection with label dependency and streaming labels (MSDS) [16], multi-label feature selection with streaming labels (MLFSL) [28], label supplementation for multi-label feature selection (LSMFS) [38], maximum label supplementation for multi-label feature selection (MLSMFS) [38], and constraint regression and adaptive spectral graph (CSASG [46]), respectively. Details of these algorithms are provided below.
MSDS: It acquires features by exploring the feature significance, label significance, and label specific features, simultaneously.LSMFS: It leverages label relationships to extract all feature supplementary information for each label from other labels.MLSMFS: It is similar to LSMFS, but it maximizes the feature supplementary information of each label from other labels.MSFS: It realizes streaming feature selection by conducting online relevance and redundancy analysis.OMNRS: It sets the bounds of pairwise correlation between features to discard redundant features.MLFSL: It is an MFS algorithm based on streaming labels, which fuses the feature rankings by minimizing the overall weighted deviation.CSASG: It proposes a multi-label feature selection framework, which incorporates a spectral graph term based on information entropy into the manifold framework.

For the proposed method, the size of the sliding window |SW| is set to 15 in this paper. For the algorithms that obtain the feature subset, e.g., MSDS, MSFS, and OMNRS, we use the feature subset obtained by these algorithms to construct new data for prediction. For the algorithms that obtain feature ranking, e.g., MLFSL, LSMFS, MLSMFS, and CSASG, the first *p* features are selected, which depends on the dimension of the feature subset obtained by the OSLGC algorithm. Furthermore, we select the average precision (AP), Hamming loss (HL), one error (OE), and macro-F1 (F1), as the evaluation metrics. Due to space limitations, information on these metrics will not be provided in detail. The formulas and descriptions of all the evaluation metrics are provided in [47,48]. Finally, MLkNN (*k* = 10) is selected as the basic classifier.

### 5.3. Experimental Results

Table 4, Table 5, Table 6 and Table 7 display the results for the different evaluation metrics, where the symbol “↓ (↑)” indicates “the smaller (larger), the better”. Boldface highlights the best prediction performance for a specific dataset, and the penultimate row in each table shows the average value of the algorithm on all datasets. Furthermore, the Win/Draw/Loss record provides the number of datasets where OSLGC outperforms, performs equally to, and underperforms compared to the other algorithms, respectively. The experimental results indicate that OSLGC exhibits strong competitiveness compared with other algorithms; the experimental results also provide some interesting insights.
For web text data, OSLGC is capable of achieving the best predictive performance on at least 7 out of the 8 datasets on all the evaluation metrics. This suggests that the proposed method is suitable for selecting features for web text data.For the Birds and Scene data, OSLGC achieves the best result on 3 out of 4 evaluation metrics. For the remaining evaluation metric, OSLGC ranks second with a disadvantage of 0.96 % and 1.51 %, respectively. This result indicates that OSLGC can also be applied to the classification problem of other data types, such as images and audio.The average prediction results of all datasets were comprehensively investigated, with the results showing that the performance of OSLGC has obvious advantages. Furthermore, the Win/Draw/Loss records clearly demonstrate that OSLGC can outperform the other algorithms.Although MSFS, OMNRS, and OSLGC are proposed to manage streaming features, the performance advantage of OSLGC confirms that label group correlation and feature interaction can provide additional information for processing streaming features.

OSLGC is able to make use of label group correlation to guide feature selection, and adds online feature interaction analysis to provide hidden information for predictive labels. By combining the potential contributions of the feature space and the label space, OSLGC performs very competitively compared to other mature MFS methods.

**Table 4 entropy-25-01071-t004:** Results for different algorithms on Average Precision (↑).

Average Precision	MSDS	MLSMFS	LSMFS	MSFS	OMNRS	MLFSL	CSASG	OSLGC
Business	0.8748	0.8705	0.8707	0.8667	0.8746	0.8750	0.8755	**0.8782**
Computer	0.6410	0.6329	0.6328	0.6303	0.6420	0.6260	0.6397	**0.6458**
Education	0.5538	0.5515	0.5319	0.5475	0.5547	0.5478	0.5599	**0.5636**
Entertainment	0.5617	0.5487	0.5685	0.5632	0.5704	0.5626	0.5649	**0.5809**
Health	0.6883	0.6617	0.6701	0.6815	0.6894	0.6551	0.7013	**0.7040**
Recreation	0.4904	0.4628	0.4774	0.4921	0.4991	0.4459	0.4824	**0.5083**
Reference	0.6238	0.6232	0.6205	0.6252	**0.6332**	0.6170	0.6290	0.6324
Society	0.5932	0.5902	0.5698	0.5961	0.5849	0.5862	0.5976	**0.5983**
Birds	0.4877	0.4603	0.4563	0.5181	0.4842	0.4614	0.5260	**0.5317**
Scene	0.8372	0.8331	0.8331	0.6756	0.8375	0.8428	0.8430	**0.8451**
Average	0.6352	0.6235	0.6231	0.6196	0.6370	0.6220	0.6419	**0.6488**
Win/Draw/Loss	10/0/0	10/0/0	10/0/0	10/0/0	9/0/1	10/0/0	10/0/0	-

**Table 5 entropy-25-01071-t005:** Results for different algorithms on Hamming Loss (↓).

Hamming Loss	MSDS	MLSMFS	LSMFS	MSFS	OMNRS	MLFSL	CSASG	OSLGC
Business	0.0276	0.0281	0.0283	0.0283	0.0274	0.0275	0.0276	**0.0273**
Computer	0.0399	0.0404	0.0401	0.0400	0.0397	0.0417	0.0398	**0.0396**
Education	0.0400	0.0410	0.0413	0.0413	0.0408	0.0410	0.0403	**0.0398**
Entertainment	0.0616	0.0624	0.0615	0.0621	0.0617	0.0615	0.0606	**0.0600**
Health	0.0433	0.0447	0.0450	0.0434	0.0417	0.0456	0.0414	**0.0408**
Recreation	0.0607	0.0617	0.0614	0.0613	0.0595	0.0635	0.0603	**0.0593**
Reference	0.0311	0.0311	0.0291	0.0304	**0.0294**	0.0311	0.0306	0.0301
Society	0.0555	0.0559	0.0585	0.0553	0.0565	0.0575	0.0553	**0.0549**
Birds	0.0513	0.0536	0.0497	0.0484	0.0518	0.0513	0.0489	**0.0463**
Scene	0.0988	0.1002	0.1002	0.1637	0.1014	0.1019	0.1009	**0.0957**
Average	0.0510	0.0519	0.0515	0.0574	0.0510	0.0523	0.0506	**0.0494**
Win/Draw/Loss	10/0/0	10/0/0	10/0/0	10/0/0	9/0/1	10/0/0	10/0/0	-

**Table 6 entropy-25-01071-t006:** Results for different algorithms on One Error (↓).

One Error	MSDS	MLSMFS	LSMFS	MSFS	OMNRS	MLFSL	CSASG	OSLGC
Business	0.1240	0.1323	0.1323	0.1360	0.1247	0.1230	0.1233	**0.1187**
Computer	0.4273	0.4387	0.4387	0.4457	0.4330	0.4583	0.4313	**0.4197**
Education	0.5673	0.5910	0.6097	0.5893	0.5810	0.5907	0.5767	**0.5653**
Entertainment	0.5947	0.6057	0.5837	0.5910	0.5783	0.5903	0.5880	**0.5620**
Health	0.3967	0.4383	0.4140	0.4157	0.4043	0.4497	0.3860	**0.3747**
Recreation	0.6557	0.6897	0.6730	0.6517	0.6413	0.7157	0.6667	**0.6277**
Reference	0.4713	0.4843	0.4710	0.4697	**0.4527**	0.4840	0.4613	0.4650
Society	0.4500	0.4587	0.4780	0.4473	0.4613	0.4677	0.4453	**0.4427**
Birds	0.6279	0.6686	0.6512	0.5581	0.6221	0.6628	**0.5349**	0.5465
Scene	0.2676	0.2742	0.2742	0.5084	0.2667	0.2567	0.2550	**0.2525**
Average	0.4582	0.4782	0.4726	0.4813	0.4565	0.4799	0.4469	**0.4375**
Win/Draw/Loss	10/0/0	10/0/0	10/0/0	10/0/0	9/0/1	10/0/0	9/0/1	-

**Table 7 entropy-25-01071-t007:** Results for different algorithms on Macro_F (↑).

Macro_F	MSDS	MLSMFS	LSMFS	MSFS	OMNRS	MLFSL	CSASG	OSLGC
Business	0.1602	0.1387	0.1326	0.1210	0.0852	0.1465	0.1591	**0.1668**
Computer	0.0996	0.0902	0.0703	0.0612	0.0714	0.0727	0.0910	**0.0919**
Education	0.1329	**0.1347**	0.1088	0.1326	0.0786	0.1312	0.1284	0.1288
Entertainment	0.1259	0.0960	0.1162	0.1145	0.1125	0.1168	0.1336	**0.1391**
Health	0.2270	0.1682	0.1537	0.1997	0.1623	0.1632	0.2435	**0.2261**
Recreation	0.1033	0.0854	0.0790	0.0834	0.1243	0.0556	0.1019	**0.1307**
Reference	0.1191	0.1111	0.1172	0.1126	0.0747	0.1180	0.1251	**0.1286**
Society	0.0855	0.0614	0.0346	0.0855	0.0581	0.0442	0.0771	**0.0791**
Birds	0.0766	0.0412	0.0460	0.0855	0.0434	0.0503	0.0573	**0.1129**
Scene	**0.7177**	0.6921	0.6921	0.3564	0.6859	0.7019	0.6834	0.7026
Average	0.1848	0.1619	0.1550	0.1352	0.1496	0.1600	0.1800	**0.1906**
Win/Draw/Loss	9/0/1	9/0/1	10/0/0	10/0/0	10/0/0	10/0/0	10/0/0	-

#### 5.3.1. Statistical Tests

To assess the statistical significance of the observed differences between the eight algorithms, we used the Friedman test [49]. The Friedman test ranks the prediction performance obtained by each dataset. The best algorithm ranks first, the sub-optimal algorithm ranks second, and so on. For *K* algorithms and *N* datasets, $r_{j}^{i}$ represents the rank of the *i* algorithm on the *j* dataset, and $R_{i}=1/N\sum_{j=1}^{N}r_{j}^{i}$ represents the average rank of the *i* algorithm. Under the null hypothesis (i.e., all algorithms are equivalent), the Friedman statistic $F_{F}$ obeys the Fisher distribution of degrees of freedom (K−1) and (K−1)(N−1):(17)$$\;F_{F}=\frac{\left(N-1\right)\chi_{F}^{2}}{N\left(K-1\right)-\chi_{F}^{2}},\;\mathrm{where}\;\chi_{F}^{2}=\frac{12N}{K(K+1)}\left(\sum_{i=1}^{K}R_{i}^{2}-\frac{K{\left(K+1\right)}^{2}}{4}\right).\;$$

Table 8 summarizes the value of $F_{F}$ and the corresponding critical value. Based on the Friedman test, the null hypothesis is rejected at a significance level of 0.10. Consequently, it is necessary to use a post hoc test to further analyze the relative performance between the algorithms. As the experiments focus on the performance difference between OSLGC and other algorithms, we chose the Bonferroni–Dunn test [50] to serve this purpose. In this test, the performance difference between OSLGC and one comparison algorithm is compared using the critical difference (CD), and $CD_{\alpha }=q_{\alpha }\cdot \sqrt{\frac{K(K+1)}{6N}}$, where $q_{\alpha }=2.450$ at α = 0.10; thus, we can compute $CD_{0.1}=2.6838$.

Figure 3 gives the CD diagrams, where the average rank of each algorithm is plotted on the coordinate axis. The best performance rank is on the rightmost side of the coordinate axis, and conversely, the worst rank is on the leftmost side of the coordinate axis. In each subfigure, if the average rank between OSLGC and one comparison algorithm is connected by a CD line, it indicates that the performance of the two algorithms is comparable and statistically indistinguishable. Otherwise, if the average rank of a comparison algorithm is outside a CD, it is considered to have a significantly different performance from OSLGC.

From Figure 3, we can observe that: (1) OSLGC has obvious advantages over LSMFS, MLSMFS, MLFSL, and MSFS with respect to all the evaluation metrics; (2) OSLGC achieves comparable performance with CSASG for each evaluation metric, but, different from the setting of the known static feature space of CSASG, OSLGC selects features by assuming the dynamic arrival of features, which entails a process of selecting the best feature with local feature information; (3) It is noteworthy that, although OSLGC cannot be significantly distinguished from all the algorithms, OSLGC exhibits significant advantages over the other feature selection algorithms. In summary, OSLGC exhibits a stronger statistical performance than LSMFS, MLSMFS, MLFSL, MSFS, MSDS, OMNRS, and CSASG.

#### 5.3.2. Stability Analysis

In this subsection, we employ spiderweb plots to verify the stability of the algorithms. Because the results generated by the algorithm on different evaluation metrics are quite different, to reasonably compare, we standardize the prediction results to a standard range [0.1, 0.5]. The spiderweb diagram has the following characteristics: (1) The larger the area surrounded by the same color line, the better the performance and the stability of the algorithm. (2) The closer the normalized value is to 0.5, the better the performance. (3) The closer the shape of the encircling line corresponding to the algorithm is to a regular polygon, the better the stability of the algorithm. Figure 4 shows spider diagrams for all the evaluation metrics, where each corner denotes a dataset and different colored lines represent different MFS algorithms, respectively.

By analyzing Figure 4, it is found that: (1) Among all the algorithms, the area surrounded by OSLGC is the largest, which indicates that OSLGC has the best performance; (2) The polygon enclosed by OSLGC is approximately a regular polygon with respect to the average precision and macro-F1. This indicates that the performance obtained by OSLGC is relatively stable on different datasets; (3) Furthermore, although the polygon enclosed by OSLGC is not a regular polygon with respect to the Hamming loss and one error metrics, the fluctuation range of OSLGC at each vertex is relatively small. In summary, compared with the other algorithms, the OSLGC algorithm has obvious advantages in terms of performance and stability.

#### 5.3.3. Ablation Experiment

To evaluate the contribution of the label group correlation, we conducted an ablation empirical study by removing the label group correlation in Algorithm 1 and derived a variant of the OSLGC algorithm, called the OSLGC-RLC algorithm. Table 9 displays the results for OSLGC and OSLGC-RLC. Due to space limitations, we select three datasets for experimental verification: Recreation, Entertainment, and Social. Considering the results in Table 9, it is observed that OSLGC significantly outperforms OSLGC-RLC on all the evaluation metrics. In conclusion, the above results suggest that considering the label group correlation is an effective strategy in feature selection.

## 6. Conclusions

In this paper, we have presented a new online multi-label streaming feature selection method, called OSLGC, to select relevant or interactive features from streaming features. In OSLGC, a set of trees is constructed using graph theory that is able to divide strongly related labels into the same tree, and which applies a streaming feature selection strategy using sliding windows, which identifies the relevant, interactive, and irrelevant features in an online manner. OSLGC can be divided into two parts: online feature relevance analysis and online feature interaction analysis. For online feature relevance analysis, we designed the feature relevance terms to provide a basis for decision-making, such as for selection, delay, and abandonment. For online feature interaction analysis, we defined an enhanced feature relevance item that prefers to select a group of interactive features from the delay decisions corresponding to the online relevance analysis. Based on experiments undertaken, our research showed that OSLGC achieved a high level of competitive performance against other advanced competitors.

In future work, we intend to combine label-specific features and common features to design streaming feature selection strategies. Furthermore, we are committed to building streaming feature selection strategies that are suitable for large-scale data.

## Figures and Tables

**Figure 1 entropy-25-01071-f001:**
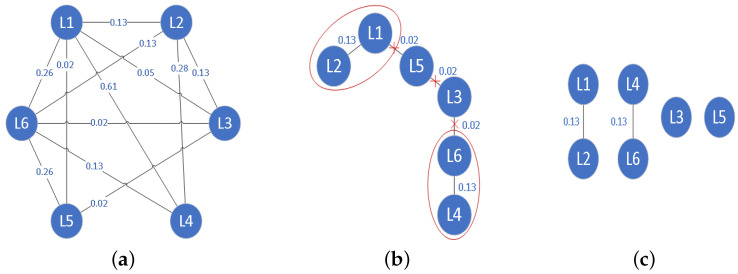
The relationship among labels. (**a**) label correlation matrix, (**b**) minimum spanning tree, and (**c**) label groups.

**Figure 3 entropy-25-01071-f003:**
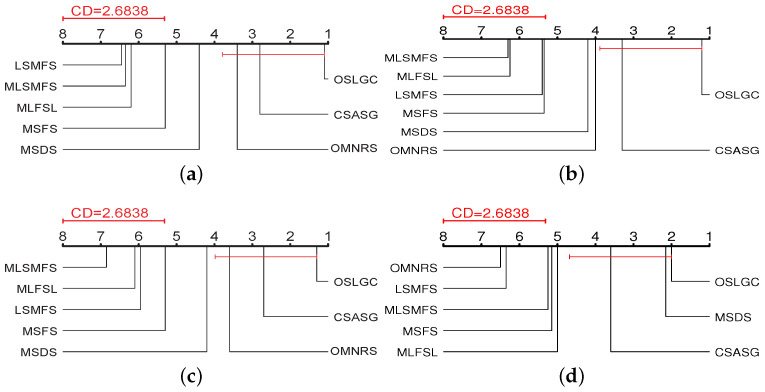
The CD diagrams using the Bonferroni–Dunn test. (**a**) Average precision, (**b**) Hamming loss, (**c**) One error, and (**d**) Macro-F1.

**Figure 4 entropy-25-01071-f004:**
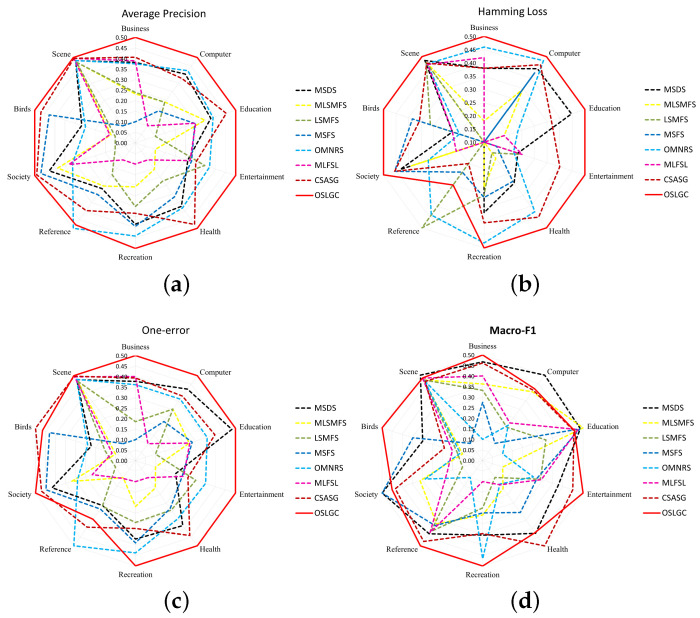
Spiderweb diagrams showing the stability of the algorithm. (**a**) Average precision, (**b**) Hamming loss, (**c**) One error, and (**d**) Macro-F1.

**Table 1 entropy-25-01071-t001:** Example of multi-label data.

Instance	$f_{1}$	$f_{2}$	$l_{1}$	$l_{2}$	$l_{3}$	$l_{4}$	$l_{5}$	$l_{6}$
$x_{1}$	1	0	+1	−1	+1	+ 1	−1	+1
$x_{2}$	1	0	+1	−1	+1	+1	−1	+ 1
$x_{3}$	0	1	−1	+1	−1	−1	+1	−1
$x_{4}$	0	1	−1	+1	+1	−1	+1	−1
$x_{5}$	1	1	−1	−1	+1	−1	−1	+1
$x_{6}$	1	0	+1	−1	−1	+1	+1	−1
$x_{7}$	0	1	+1	+1	−1	−1	−1	+1
$x_{8}$	0	1	−1	+1	+1	−1	−1	−1
$x_{9}$	1	0	+1	+1	−1	+1	−1	+1
$x_{10}$	1	1	+1	−1	+1	+1	+ 1	+1

**Table 3 entropy-25-01071-t003:** Detailed description of datasets.

Dataset	|S|	F(S)	L(S)	LCard(S)	LDen(S)	Domain
Business	5000	438	30	1.599	0.053	Text
Computer	5000	681	33	1.507	0.046	Text
Education	5000	550	33	1.463	0.044	Text
Entertainment	5000	640	21	1.414	0.067	Text
Health	5000	612	32	1.662	0.052	Text
Recreation	5000	606	22	1.423	0.065	Text
Reference	5000	793	33	1.169	0.035	Text
Society	5000	636	27	1.67	0.062	Text
Birds	645	260	19	1.014	0.053	Audio
Scene	2407	294	6	1.074	0.179	Image

**Table 8 entropy-25-01071-t008:** Friedman statistics $F_{F}$ and critical value.

Evaluation Metric	$F_{F}$	Critical Value (α=0.10)
Average Precision	15.2697	1.74
Hamming Loss	9.0301	
One Error	13.5067	
Macro-F1	9.2081	

**Table 9 entropy-25-01071-t009:** Results between OSLGC and OSLGC-RLC.

Evaluation Metric	Recreation		Entertainment		Social	
	OSLGC	OSLGC-RLC	OSLGC	OSLGC-RLC	OSLGC	OSLGC-RLC
Average Precision	**0.5083**	0.4996	**0.5809**	0.5713	**0.7126**	0.7036
Hamming Loss	**0.0593**	0.0601	**0.0600**	0.0603	**0.0250**	0.0252
One Error	**0.6277**	0.6440	**0.5620**	0.5783	**0.3767**	0.3957
Macro-F1	**0.1307**	0.1129	**0.1391**	0.1375	**0.1500**	0.1400

## Data Availability

Real-world datasets that we use in our experiments are publicly available. These data can be found here: http://www.lamda.nju.edu.cn/code_MDDM.ashx and https://mulan.sourceforge.net/datasets-mlc.html.

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
