# Peer review of "Online Multi-Label Streaming Feature Selection Based on Label Group Correlation and Feature Interaction"

_entropy, 2023, doi:10.3390/e25071071_

Round 1

Reviewer 1 Report

This paper presented a novel online streaming feature selection based on label group correlation and feature interaction. Experiments were conducted to evaluate the effectiveness of the proposed method. Overall, the paper is high-quality and well-organized. I have some minor comments: 

(1) Please report more specific performance of the experiment results in the abstract. 

(2) There are a lot of abbreviations in this paper, all the abbreviations need to be spelled out during the first it was presented. 

(3) The last two datasets (Birds and Scene) are much more smaller than the other datasets, I am wondering how this will affect the performance and the conclusion. 

(4) Figure 4 has some issues (when taking the screenshot). 

I suggest the authors using professional language editing before the paper being published. 

Reviewer 2 Report

Manuscript Number: entropy-2471807

Title:  Online multi-label streaming feature selection based on label group correlation and feature interaction

Article Type: Review Article

The subject of research includes in this journal. The research paper is an interesting because authors, present a novel online streaming feature selection based on label group correlation and   feature interaction. At the beginning divided the label groups with the help of graph theory. After that, they integrate label weight and mutual information to quantify   feature relevance and feature interaction for the label group. Subsequently, a novel feature selection   framework by using sliding windows is designed, which includes online feature relevance analysis   and online feature interaction analysis.

This method is already known and used widely. In the paper titled “Online multi-label streaming feature selection based on label group correlation and feature interaction”.

In my recommendation is major revision.

1.       The quality of the language is insufficient. Have a native speaker or similar assist you.

2.       Please explain why you chose these methods?

3.       Please correct the caption of Figure 1. The description of what does mean a, b, c?

4.       Please standardize the notation in the matrix to two decimal places.

5.       Please explain what does mean the acronyms HE, ME, IE in Figure 2?

6.       I would very much appreciate a more detailed description of the dataset as well as their numbers (table 3).

7.       What does mean acronym in Table 4 for instance (MSDS MLSMFS etc.)

8.       Please correct the caption of Figure 4. The description of what does mean a, b, c?

9.       In my opinion, the final conclusions should also be drawn up as they are not as specific as the scope of work?

10.    Authors should include more work in the literature review. Because after analyzing the Scopus or WoS databases you can find many similar works?

Manuscript Number: entropy-2471807

Title:  Online multi-label streaming feature selection based on label group correlation and feature interaction

Article Type: Review Article

The subject of research includes in this journal. The research paper is an interesting because authors, present a novel online streaming feature selection based on label group correlation and   feature interaction. At the beginning divided the label groups with the help of graph theory. After that, they integrate label weight and mutual information to quantify   feature relevance and feature interaction for the label group. Subsequently, a novel feature selection   framework by using sliding windows is designed, which includes online feature relevance analysis   and online feature interaction analysis.

This method is already known and used widely. In the paper titled “Online multi-label streaming feature selection based on label group correlation and feature interaction”.

In my recommendation is major revision.

1.       The quality of the language is insufficient. Have a native speaker or similar assist you.

2.       Please explain why you chose these methods?

3.       Please correct the caption of Figure 1. The description of what does mean a, b, c?

4.       Please standardize the notation in the matrix to two decimal places.

5.       Please explain what does mean the acronyms HE, ME, IE in Figure 2?

6.       I would very much appreciate a more detailed description of the dataset as well as their numbers (table 3).

7.       What does mean acronym in Table 4 for instance (MSDS MLSMFS etc.)

8.       Please correct the caption of Figure 4. The description of what does mean a, b, c?

9.       In my opinion, the final conclusions should also be drawn up as they are not as specific as the scope of work?

10.    Authors should include more work in the literature review. Because after analyzing the Scopus or WoS databases you can find many similar works?

Round 2

Reviewer 2 Report

Thank you for your answers I have no objections.